# Importance of Public Transport Networks for Reconciling the Spatial Distribution of Dengue and the Association of Socio-Economic Factors with Dengue Risk in Bangkok, Thailand

**DOI:** 10.3390/ijerph191610123

**Published:** 2022-08-16

**Authors:** Bertrand Lefebvre, Rojina Karki, Renaud Misslin, Kanchana Nakhapakorn, Eric Daudé, Richard E. Paul

**Affiliations:** 1French Institute of Pondicherry, UMIFRE 21 CNRS-MEAE, Pondicherry 605001, India; 2CNRS, ARENES—UMR 6051, EHESP, Université de Rennes, 35000 Rennes, France; 3INRAE, 68000 Colmar, France; 4Faculty of Environment and Resource Studies, Mahidol University, Salaya, Nakhon Pathom 73170, Thailand; 5CNRS, UMR 6266 IDEES, 7 rue Thomas Becket, 76821 Rouen, France; 6Institut Pasteur, Université de Paris, CNRS, UMR 2000, Unité de Génétique Fonctionnelle des Maladies Infectieuses, 75015 Paris, France

**Keywords:** dengue, socio-economic risk, spatial clusters, mobility, transport system, Bangkok

## Abstract

Dengue is the most widespread mosquito-borne viral disease of man and spreading at an alarming rate. Socio-economic inequality has long been thought to contribute to providing an environment for viral propagation. However, identifying socio-economic (SE) risk factors is confounded by intra-urban daily human mobility, with virus being ferried across cities. This study aimed to identify SE variables associated with dengue at a subdistrict level in Bangkok, analyse how they explain observed dengue hotspots and assess the impact of mobility networks on such associations. Using meteorological, dengue case, national statistics, and transport databases from the Bangkok authorities, we applied statistical association and spatial analyses to identify SE variables associated with dengue and spatial hotspots and the extent to which incorporating transport data impacts the observed associations. We identified three SE risk factors at the subdistrict level: lack of education, % of houses being cement/brick, and number of houses as being associated with increased risk of dengue. Spatial hotspots of dengue were found to occur consistently in the centre of the city, but which did not entirely have the socio-economic risk factor characteristics. Incorporation of the intra-urban transport network, however, much improved the overall statistical association of the socio-economic variables with dengue incidence and reconciled the incongruous difference between the spatial hotspots and the SE risk factors. Our study suggests that incorporating transport networks enables a more real-world analysis within urban areas and should enable improvements in the identification of risk factors.

## 1. Introduction

Dengue is a rapidly emerging mosquito-borne infection, caused by any one of four viral serotypes (denoted DENV-1, 2, 3, 4), with an increase in incidence of thirty-fold over the last 50 years [1]. The disease is endemic in over 100 countries [2,3]. More than 3.5 billion people are at risk of DENV infection and recent estimates suggest that there are 390 million DENV infections every year, of which 100 million cause clinical symptoms [4]. The increase in global transmission of this disease has been linked to several factors such as global trade, international travel, rapid urbanization, and ineffective vector control strategies [5,6,7,8].

Dengue virus is mainly transmitted by Aedes aegypti mosquitoes, which are widely present in tropical and subtropical areas and well-adapted to an urban environment. Dengue risk is associated with climatic factors particularly temperature and rainfall, which impact upon the mosquito abundance and vectorial capacity [6,9,10]. Urbanization has been frequently linked with the endemicity of the disease, where high population density coupled with poor environmental hygiene provide a conducive environment for mosquito vector breeding and increased probability of transmission [11,12,13]. Dengue is associated with a wide range of socio-economic (SE) factors that alter risk of exposure to infectious mosquitoes and which can vary at very local scales [11,12,13,14,15,16,17,18,19]. However, these associated risk factors are not systematic [20,21], likely influenced by the role of human mobility in ferrying the virus from places of high environmental risk throughout the city [22,23,24]. The identification of the source of infection and the subsequent socio-spatial structure of the intra-urban spread of DENV would clearly aid the local public health services to better allocate resources [25].

Bangkok, the capital city of Thailand, bears a considerable burden of reported cases of dengue and has year-round endemic transmission [26]. In Bangkok, dengue-confirmed cases from the hospitals are reported to the Bangkok Metropolitan Administration (BMA) and Ministry of Public Health (MoPH) via an online operating system “Epi-net” [27]. Based on this notification, intervention is activated, which is largely based on fumigation of insecticide in the patient’s household and the neighboring area within a radius of 100 m [27]. Some of the major challenges faced by this system have been delayed intervention, which relies solely on the clinical confirmation of cases and reporting, followed by difficulty in tracing the address of identified cases [27]. Hence, understanding the spatial and socio-economic distribution of risk associated with the disease at a subdistrict level in such an endemic setting could help assign probable risk to an area and therefore enable early vector control-targeted interventions and prevent frequent outbreaks. Within this context, understanding how intra-urban mobility in Bangkok may shape the spread of the virus is important. Bangkok has a very well developed intra-urban public transport system comprised of BTS Skytrains, MRT Subways, Airport Rail Link, buses, and ferry boats that enable rapid transport throughout the city. Geographical knowledge of the structure of the transport network can provide a proxy for human intra-urban mobility and thus enable its inclusion in understanding dengue risk.

The overall aim of this study was firstly to identify socio-economic and spatial risk factors for dengue incidence across subdistricts in Bangkok, all the while taking into account the influence of meteorological factors and, secondly, to assess the added value of incorporating intra-urban transport networks as a proxy for human mobility.

## 2. Materials and Methods

### 2.1. Study Design

This is a three-step retrospective analytical observational study to assess the association firstly of meteorological factors with monthly dengue incidence across Bangkok over a 14-year period. Secondly, association of socio-economic (SE) variables and previously identified significantly associated meteorological variables with dengue incidence over a two-year period at a sub-district level was assessed. Finally, the impact of inclusion of transport networks on the observed SE associations was assessed.

### 2.2. Study Area

Bangkok is the capital city in the central region of Thailand (latitude 100°31′, longitude 100°0′) and covers an area of 1569 km^2^. Bangkok has a tropical climate. Rainfall occurs throughout the year but with highest rainfall from May to October. Likewise, temperature fluctuates year-round, ranging from 22–35 °C on average, but can soar to 40 °C in March–May. According to the 2010 census, Bangkok had a population of 8.3 million, out of which 4.3 million were registered [28]. This population inflates up to 15 million during the day mainly due to commuters from surrounding areas. The city has an average population density of 5294.5 persons/km^2^ [28]. Bangkok is composed of 50 districts (khet), and 180 subdistricts (khwaeng) in 2019.

### 2.3. Data Sources

#### 2.3.1. Dengue Incidence Data

The data were obtained from two different sources. Dengue incidence data for the year 2000–2013 were obtained from the Bureau of Epidemiology, MoPH in the context of the FP7 DENFREE program [29]. This dataset is comprised of dengue cases per month per year in Bangkok and used to identify meteorological factors associated with dengue incidence for subsequent inclusion in analyses of socio-economic variable risk analysis as covariates. The second dataset was obtained from the Health Department of the Bangkok Metropolitan Administration for 2012–2013, which consisted of individual case level data on a monthly basis, which were mapped with certainty to the subdistrict level and are the outcome of interest in this study.

#### 2.3.2. Environmental Data

Meteorological data (temperature and precipitation) for the year 2000–2012 were obtained from the Bangkok Meteorological Department. For the year 2013, data were retrieved from Don Muang Airport meteorological station records, which were then merged with the former, being found to be consistent over previous years. Daily values of the maximum and minimum temperature and mean and maximum precipitation were averaged at a monthly level for each year. For temperature, two new variables, Diurnal Temperature Range (DTR) and mean temperature were generated given their known influence on dengue incidence [7,30]. In addition to concurrent monthly values, time lags of 1, 2, and 3 months were assessed for lag effects of these meteorological variables on the disease incidence fitted with year. Incorporation of the effect of year on dengue incidence attempted to take into account other factors, such as herd immunity, differing viral serotypes, etc., which might confound the underlying effect of the environmental variables.

#### 2.3.3. Socio-Economic Variables

The dataset with socio-economic (SE) variables was obtained from the 2010 census data conducted by the National Statistical Office of Thailand [28]. This dataset included subdistrict level aggregated data for Bangkok on a wide range of SE factors that included education, nationality/immigration, occupation, type and structure of housing, type of water sources, different household amenities (e.g., refrigerator, air conditioner, washing machine, TV, car, etc.), subdistrict surface area (km^2^), number of households, population characteristics (age, sex, occupation, education level). All these variables were measured in relative percentages except area, number of houses, and population density, which were measured in numbers. Based on literature review and/or biological relevance, only those variables considered to have pertinent effects on dengue were selected for further analysis. These selected variables and their definitions are presented in Table A1. A brief rationale of each selected variable is as follows:(1)Lack of education can lead to precarious working/housing conditions as it relates to household income, thereby increasing dengue risk [2,3];(2)Immigration can impose a significant dengue risk due to lack of access to health care or proper housing conditions [31];(3)Agricultural areas generally have higher rainfall and humidity, lower abundance of Ae. aegypti but higher of Aedes albopictus [32];(4)Manual work sites such as construction sites are found to be potential areas of dengue clusters [33];(5)High population density is a known risk factor for dengue transmission [34];(6)Age will reflect differential exposure to DENV and subsequent level of acquired immunity and thus could be a potential confounder for dengue incidence [31];(7)Construction materials for houses have been identified as a risk factor in the past; for example, cement/brick houses have lower temperature and high humidity, which are favorable for adult mosquito survival [35,36];(8)Shop houses usually have longer hours with windows and doors are open and thus provide easy entrance for mosquitoes [35];(9)Different types of water sources were considered as they provide effective breeding sites for mosquitoes and thus promote vector density [37,38];(10)Pit toilets could be potential breeding sites for mosquitoes and thus could increase dengue risk [39,40];(11)Air conditioners promote indoor breeding sites and impact survival of Ae. aegypti mosquito through maintenance of clement temperatures;(12)Land surface coverage (vegetation, road, water bodies, extent built-up) will determine the ecological suitability for mosquito abundance/survival and is also a proxy for human population density [12,13].

#### 2.3.4. Intra-Urban Transport Networks Variables

In order to take into account the impact of public transport and human mobility in the spatial diffusion of dengue [41], bus, metro and ferry stops (N = 3767), as well as the lines (N = 209) that make up Bangkok’s public transport network were incorporated into a GIS from several sources (Bangkok Metropolitan Administration, OpenStreetMap, TransitBangkok). Road networks were also added for comparison. Several indicators were derived from these datasets at the subdistrict level (number of public transport stops, public transport stops density, total road length, road density) to measure the level of accessibility and of transport infrastructure of each subdistrict.

### 2.4. Statistical Methods

#### 2.4.1. Association Analyses

To assess the association of meteorological variables with time lags (lag 0 to 3 months) with monthly dengue incidence over the period 2000–2013 across all Bangkok, we fitted Generalized Loglinear Models (GLMs). In order to avoid overfitting of too many “alike” variables differing only in the lag time, the lag time of each variable in a univariate analysis that gave the best fit was selected for the multivariable analysis. Thus, for each meteorological variable (mean, minimum, maximum temperature, DTR and mean, max rainfall), the time lag that explained the largest percentage of variation in the dengue incidence data was selected for a multivariate analysis with backward elimination of non-significant variables until a final adequate model was achieved.

To ensure comparability of the datasets over the study period (2000–2013), data were aggregated across the 160 subdistricts that made up the Bangkok Metropolitan Administration prior to the 2009 administrative reform. To assess the association of socio-economic variables with dengue incidence data from 2012 and 2013 localized to subdistrict level, we performed univariate analyses, fitting Generalized Loglinear Mixed Models (GLMMs) with Log(e) population fitted as an offset and subdistrict as the random variable to account for the repeated (monthly) measures of dengue within each subdistrict. All variables for which the univariate analyses yielded a *p* < 0.2 were retained and fitted in the multivariable analysis, including the identified associated meteorological variables above, and a final adequate model including only significant variables was achieved by backward elimination of non-significant variables. For the multivariable analyses, an adjusted Relative Risk (aRR) was calculated to show the direction and size of the association of the explanatory variables with the output variables. Relative Risk for the case of continuous explanatory variables is the percentage increase in dengue cases for a unit increase in the explanatory variable (e.g., 1 °C for temperature, 1 mm for precipitation, 1% for the %SE variables, etc.). The aRR is the relative risk of a variable adjusted for all the other co-fitted variables in the multivariable analysis.

Because many statistical tests were performed, for interpretation of statistically significant variables, we applied Bonferroni’s correction to yield a corrected p-value significance threshold. Thus, only those variables lower than the corrected p-value threshold were considered of significant interest.

In order to assess whether the dengue clusters (hotspots) identified through Local Moran’s Statistic (LISA) (see below) were associated with the significant SE variables identified above, we fitted a GLM with normal error structure and cluster type as an explanatory factor. However, because some of the SE variables were percentages, we had to arcsine transform them prior to the analysis.

To take into account connectivity between subdistricts, we retrieved information on bus, metro (BTS, MRT, Airport Link), and ferry lines and counted the number of lines connecting bus stands, metro stations ferry piers from one subdistrict to another and within a subdistrict, generating a symmetric transport matrix of connectivity among all subdistricts. From this transport matrix, we then generated a new matrix to quantify the similarity in the extent of connectedness of a subdistrict as compared to that of all the other subdistricts. To achieve this, as discussed in detail in [42,43], when there are n individuals (here subdistricts) distributed in Euclidean space, an n × n similarity matrix can be generated, whereby each column represents a subdistrict and contains the number of connections with the subdistricts arranged as rows. The similarity, S, of a subdistrict to another, is calculated by comparing the number of connections each has to all subdistricts (including itself) taking into account the range in the numbers of connections any subdistrict has. A pair of subdistricts having exactly the same number of connections to all subdistricts will have a similarity value of 1. The Euclidean similarity matrix uses the following formula: S = 1 − [(x_i_ − x_j_)/range of x_k_]^2^ where x is the number of transport connections of the ith and jth subdistrict to the kth subdistrict and the range is the maximum-minimum number of connections that occur from the kth subdistrict to all subdistricts [42,43]. The similarity value, S, of the ith and jth subdistrict is then the average of all the values computed to all the subdistricts. The value of similarity then takes any value from 0 to 1. We also measured the (Euclidean) distance between the geographic centre of all pairs of subdistricts and calculated a distance Similarity matrix. These similarity matrices were then used as weighting matrices for a second series of analyses on the association of pertinent SE variables with High/Low clusters as well as a comparative analysis of the pertinent SE variables (significant in the final multivariate analysis above) with and without the transport and distance matrices. For these analyses, we first regressed out the significant meteorological variables and year, and then took the mean residual dengue incidence per subdistrict for analysis with the SE variables with and without weighting with the transport or distance matrices.

In all analyses, a dispersion factor was estimated to account for any over-dispersion in the data. All analyses were conducted in Genstat vers. 15 (VSN International, Hemel Hempstead, UK) [44].

#### 2.4.2. Spatial Analysis

GeoDa (version 1.12.1.161, Center for Spatial Data Science, University of Chicago, USA) [45] and QGIS (version 2.18.27,QGIS Association) [46] were used for mapping the spatial distribution and further to identify spatial autocorrelation of dengue cases between various subdistricts in Bangkok. The geographical layers of Bangkok were obtained from the BMA. QGIS was used to develop dengue risk maps and layers based on the reported cases of dengue for the year 2012 and 2013 across subdistricts in Bangkok. These layers were then used in GeoDa to identify global, local spatial autocorrelation and generate a spatial correlogram.

As incidence can vary depending on the population size, with highly populated subdistricts tending to have more cases and vice versa, incidence rates were used as an outcome variable in this analysis. Dengue incidence rates were calculated based on the number of cases per subdistrict divided by their respective population size (per 10,000 population). As a measure of global spatial autocorrelation, Moran’s I was calculated for each year. This was calculated based on the spatial weight matrix that defined neighbours for each subdistrict based on the queen contiguity, i.e., common edge and a common vertex between geographical units. Random 999 permutations were applied to assess significance of global autocorrelation. Likewise, univariate Local Moran’s statistic measure (LISA) was used to identify significant local clusters and spatial outliers among subdistricts. Significance and LISA cluster maps were obtained for each year independently (for more information on the LISA methodology, see https://geodacenter.github.io/workbook/6a_local_auto/lab6a.html, accessed on 1 April 2018). Spatial autocorrelation of dengue was investigated as a function of distance through a spatial correlogram. Euclidean distance was chosen as a distance metric. Since the autocorrelations are normally sparser at longer distance, maximum distance was set to 9 km. Maximum distance implies the exact maximum distance between two subdistrict centroids. The distance interval of bins was set to 365 m.

## 3. Results

### 3.1. Impact of Meteorological Variables on the Dengue Incidence in Bangkok 2000–2013

There were 124,820 cumulative dengue cases reported during the years 2000–2013, where around 66% of cases (N = 81,760) was reported from June to November, coinciding with the monsoon season in Bangkok. The overall mean dengue incidence per month was 743, and the range of number of cases reported per month was 108 to 2260 (Figure A1). Overall, the trend of reported dengue cases fluctuated over the years, with the highest incidence reported in the year 2001 at 14,686 followed by 2013 (N = 14,134), 2008 (N = 11,846), 2010 (N = 10,912), 2011(N = 10,574), 2012 (N = 10,030).

Over the period 2000–2013, the descriptive statistics of the meteorological variables used in the association analyses were as follows: the mean of the monthly minimum temperatures was 25.5 °C (Standard deviation (SD) 1.36, Range 20.8–28.4); the mean of the monthly maximum temperatures was 33.7 °C (SD 1.15, Range 30.5–37.0); the mean of the monthly mean temperatures was 29.6 °C (SD 1.17, Range 26.1–32.7); the mean of the monthly mean DTR was 8.2 °C (SD 0.92, Range 6.3–11.0); the mean of the monthly maximum precipitation was 41.32 mm (SD 33.5, Range 0–216.8); the mean of the monthly mean precipitation was 4.75 mm (SD 4.33, Range 0–20.04). Diurnal temperature range (lag 1), maximum temperature, mean temperature (lag 3), minimum temperature (lag 3), mean precipitation (lag 1), and maximum precipitation (lag 1) gave the best fit models for their respective variables for dengue incidence, explaining 28.6%, 11.7%, 16.3%, 21.2%,17.4%, and 8.3% of variance, respectively (Table A2). Including these best-fit models in the multivariable regression, minimum temperature (lag 3) and mean temperature (lag 3) were no longer significant. Mean precipitation (lag 1) was positively associated with dengue incidence rate (*p* < 0.005) and diurnal temperature (lag 1) was negatively associated (*p* < 0.001). Maximum temperature for the same month was marginally associated with dengue incidence rate (*p* = 0.032) (Table 1); the maximum precipitation (lag 1) neared significance (*p* = 0.053). The relationship between these significant variables in relation to dengue incidence can be seen in Figure A2.

Since many statistical tests were performed a Bonferroni correction was applied, yielding a new *p*-value threshold of 0.0017 calculated as alpha/number of tests = 0.05/29. Using this value as the significance threshold, the maximum temperature was no longer considered significant for inclusion in subsequent analyses with SE variables for 2012–2013.

### 3.2. Association of Socio-Economic Factors with Dengue Incidence

The mean percentage and standard deviation of the SE variables at the level of Bangkok are shown in Table 2. The association analysis of these variables with dengue case data was carried out per subdistrict over two years (2012–2013). Seasonality has a strong role in dengue transmission in endemic settings [47] and, thus, seasons were initially analysed separately; January–June was considered as the dry season and July–December as the wet season. Univariate analyses revealed a large number of significant SE variables and which differed between seasons (Table A3). In the final minimum adequate model in the multivariable analysis, SE variables associated with dengue cases differed between seasons. In the dry season, “%Groundwater” was negatively associated with dengue cases, whereas “%Manual labor” increased risk. In the wet season, “%Cement houses” and “%No education” both increased risk (Table A4). Diurnal temperature range (lag 1) was only significantly negatively associated with dengue risk in the wet season. The strongest SE association with dengue cases was with “Nb of houses (100s)”, which was positively associated with dengue risk in both seasons. Combining seasons gave similar results to the wet season, likely because of the larger number of dengue cases occurring in that season, although “%Manual labor” and “%Groundwater” were no longer found to be associated with dengue incidence (Table 3). Overall, the meteorological variables and yearly variation had a stronger association than SE variables, with the exception of “Nb of houses (100s)” (Table 3).

### 3.3. Spatial Analysis of Dengue Incidence

The aim of this analysis was to identify any pattern of spatial autocorrelation of reported dengue across subdistricts and to identify areas of dengue hotspots, cold spots, and spatial outliers across subdistricts in Bangkok. The mean incidence rate for 2012 was 16.93/10,000 population (range: 2.32–40.35/10,000 population), whereas the mean incidence for 2013 was 20.81/10,000 population (range: 6.17–136.84/10,000 population). The spatial distribution of dengue was mapped for each year based on the incidence rates. These rates were categorized into quantiles so that each class had the same number of subdistricts and were comparable between years. Figure 1 shows the considerable inter-annual variation in the spatial distribution of dengue cases. However, for both years, high incidence rates (IRs) were concentrated in the city centre, along with areas in the eastern and western periphery and low IRs in the northern subdistricts. Global Moran’s Index for 2012 was 0.24, denoting a weak positive spatial autocorrelation (*p* = 0.001, z = 5.59 under 999 permutations). This suggests that in Bangkok, there was statistically significant clustering of dengue in subdistricts in the given year, but at a low level, suggesting some spatial dispersion as well. In 2013, Global Moran’s I was even lower at 0.12 (*p* = 0.006, z = 3.04, under 999 permutations). It is notable that 2013 had low spatial autocorrelation even though the number of reported cases was higher.

In order to further assess the clustering pattern at the subdistrict level, Local Indicators of Spatial Association (LISA) were calculated. This measure categorizes significant areas into four different cluster types. Areas with a significantly higher than mean number IR surrounded by similar areas are known as High–High clusters or hotspots, and areas with significantly lower than the mean IR surrounded by similar areas are called as Low-Low clusters or cold spots. Likewise, if areas with low cases were surrounded by areas with high IR and vice versa, they are categorized as Low–High clusters and High-Low clusters, respectively, which are also called spatial outliers. LISA analysis identified significant clusters in 40 subdistricts in 2012. High–High clusters were found in 13 subdistricts, all located at the heart of Bangkok (Figure 2a). Low–Low clusters were found in 23 subdistricts, mostly in the eastern and northern subdistricts. There were four subdistricts with Low–High clusters. These spatial outliers were detected in the city centre adjacent to the hotspot areas. No High–Low outliers were found in 2012. In 2013, 30 amongst 160 subdistricts showed significant clusters, out of which High–High clusters were identified in eight subdistricts, Low–Low clusters in 12 subdistricts, Low–High and High–Low clustering each in 5 five districts (Figure 2b). High–High clusters were again found at the city centre, along with Low–High clusters. Seven of the Low–Low clusters were the same as the previous year. High–Low clusters detected this year were found mainly in subdistricts identified as Low–Low clusters the previous year.

Over the two years, 51 out of a total of 160 subdistricts were classified into a cluster type. Nineteen subdistricts were identified as clusters/outliers in both years, 14 of which had the same category (4 High–High, 7 Low–Low and 3 Low–High), two changed from High–High to Low–High, two from Low–Low to High–Low and one from Low–High to High–High.

Spatial autocorrelation of dengue incidence rates was then investigated as a function of distance through a spatial correlogram. The number of subdistrict pairs per distance band ranged from 129 to 720, with 10,812 pairs in total. A very high positive spatial autocorrelation of 0.93 was found at a very local scale in the distance band of 0–1200 foot (365.8 m) for the year 2012 (Figure A3). For 2013, there was a weaker correlation of 0.21 observed at this distance band, despite this distance having the highest positive autocorrelation for 2013. Proximity therefore had an influence on the spatial distribution of dengue cases but only at a very local scale.

We then addressed the extent to which the observed hot and cold spot clusters had SE characteristics associated with increased risk of dengue. Only “%No education” was significantly higher in High–High clusters and cluster type explained 13% and 11% of the variation in this SE variable in 2012 and 2013, respectively (Table 4). By comparison, cluster type only explained 1% of variation in “%Cement houses” in either year, with no notable variation among cluster categories. By contrast, “Nb of houses (in 100s)” were higher in the Low–Low clusters and cluster type explained 12% and 3% of the variation in this SE variable in 2012 and 2013, respectively. Overall, the Low–High clusters were similar to High-High clusters for the SE variables. Likewise, High–Low clusters tended to have characteristics more similar to Low–Low clusters. We then combined the 2012 and 2013 clusters, retaining only those that remained in the same cluster category in both years. “%Cement houses” was significantly different between High–High and Low–Low clusters, being higher in the Low–Low clusters (t = 2.75, *p* = 0.019). In contrast to “%Cement houses”, where 31% of the variation was explained by cluster type, only 1% and 4% of variation in “%No education” and “Nb of houses (in 100s)”, respectively, was explained by cluster type.

### 3.4. Impact of Transport and Distance

As evident from Figure 2a,b, the High–High clusters tend to be very centrally located in both years and the Low–Low clusters more peripherally located. This centrality is mirrored in the transport network with peripheral areas being more poorly connected than the central areas (Figure 3). The public transport stops density is significantly higher in the High–High areas (Table 4). Cluster type explained 20% and 17% of the variation in transport density in 2012 and 2013, respectively, larger than for the SE variables as described above. When re-analysed using only those clusters that remained of the same type in both years, cluster type explained 73% of the variation in transport density. Furthermore, inclusion of public transport stops (number of bus stands, ferry piers, metro/BTS stations) per subdistrict in the multivariable risk factor model did reveal a small but significant positive association with dengue cases (Table 3).

In light of the strong association of hotspots with transport density, we then constructed a transport matrix that totalled the number of connections through public transport lanes among all subdistricts and a distance matrix between the barycentres of all subdistricts. These matrices were then used as weight matrices for statistical analyses of the association of SE variables with dengue incidence after having regressed out the year and meteorological effects (thus using the mean residual dengue case values per subdistrict) and of LISA clusters with SE variables. Incorporating either the transport or the distance similarity matrix as a weighting matrix led to a vastly improved model fit of the association of SE variables with mean dengue residuals (with transport matrix, 92% of the variance explained vs. 61% without; with the distance matrix, 96% of the variance explained). Notably the contribution of “% No education” increased from 2% to 37% with inclusion of the transport matrix and to 73% with inclusion of the distance matrix. By contrast, the contribution of “Nb of houses (in 100s)” decreased from 47% to 19% and to 23% following inclusion of the transport and distance matrices, respectively. In contrast to the similar directional effects of incorporating either matrix for “% No education” and “Nb of houses (in 100s)”, the percentage of variation explained by “%Cement housing” increased upon weighting with the transport matrix, but decreased with the distance matrix (Table 5). We also used the mean number of dengue cases per subdistrict as an explanatory variable alone, again with and without the matrix weightings. As expected, even without the matrices, the mean number of dengue cases explained a very large percentage of variation in mean residual number of dengue cases (88%); this increased to 94% after weighting with either the transport or distance matrices.

Re-analysis of the LISA High/Low dengue clusters but with inclusion of the transport weighting matrix led again to a consistent (both years) positive association of “% No education” with High–High areas and negative association with Low–Low areas as compared to uncategorized areas. There were no other consistent associations with either “Nb of houses (in 100s)” or “%cement/brick housing” for any LISA categories (Table A5).

## 4. Discussion

This study attempted to identify key socio-economic and spatial risk factors for dengue in the urban setting of Bangkok at the spatial scale of subdistrict, all the while taking into account the influence of identified meteorological factors. The major findings of this work were that although three SE variables (“%No education”, “%cement house structure”, and “Nb of houses (100s)”) were associated with increased dengue IR, they poorly explained the observed dengue hotspots. Moreover, inclusion of the intra-urban transport connectivity matrix among subdistricts vastly improved the explanatory power of the fitted model and radically altered the explanatory power of the three SE variables.

In more detail, in addition to the three aforementioned SE variables, “%manual labour” was found to be associated with increased incidence of dengue and “%groundwater use” was found to be protective and only during the dry season. This result is consistent with a reduced need to stock water afforded by well-water availability and thus reduced breeding sites associated with stocking water as observed previously [48,49]. Likewise, construction sites are known to generate potential mosquito breeding and transmission sites for the dengue virus [33,36]. The observed seasonal effect here may be a consequence of a general increase in breeding sites everywhere in the wet season, thus diluting any specific effects of groundwater and manual labour. A high percentage of lack of education was also found to be a risk factor and likely reflects poor income as well as lack of knowledge on personal protection and environmental hygiene, raising the question of whether dengue is a disease of poverty [50]. Increased risk associated with cement houses has been noted previously, potentially by providing an environment conducive for mosquito survival [35,36]. It may also reflect increased housing density as this variable was strongly positively correlated with Nb of houses (r = 0.44, *p* < 0.001).

The only strongly, positively associated SE risk factor was the “Nb of houses”, which likely reflects population density, thereby increasing both the facility for disease spread and the number of mosquito breeding sites [12]. The relationship of the latter with population density is thought to be nonlinear [51], perhaps explaining why number of houses rather than population density per se was such a strong risk factor; a larger number of houses may generate larger numbers of breeding sites independently of the human population size (e.g., flower pots in front yards and backyards).

However, overall, the SE risk factors at the subdistrict level explained very little of the variation in dengue incidence as compared to the meteorological factors at the city level, i.e., diurnal temperature range (DTR) and mean precipitation, both one month previously. Large changes in daily temperature (i.e., DTR) had impact on vectorial capacity, affecting mosquito population densities through diminished larval development and increased adult mortality, as well as modifying the susceptibility of the mosquito to viral development [9,30,52]. The negative association of DTR with dengue incidence was indeed confirmed in this study. Likewise, rainfall is directly associated with increase in natural breeding sites and thus larval density [10]. However, heavy rainfall is known to have nonlinear effects on mosquito density both due to flushing of larva and adult mortality [10]. The paucity of excessive rain during this study likely led to no evidence of any non-linearity in the relationship with dengue incidence.

The spatial analyses revealed that clustering occurs at a very local scale (<~350 m) as has been shown previously [13,53,54], likely reflecting the low dispersal of the mosquito vector from the human source of infection. This would thus likely inflate the number of local cases (within the subdistrict) and bias results based on SE status. Indeed, the dengue hotspots and cold spots did not, overall, reflect the characterized SE risk variables, with the exception of “%No education”. In light of the “forest-fire” nature of dengue clusters, there are clearly two processes at work: firstly, the probability of having sufficient dengue cases seeding an area, and secondly, the subsequent expansion of dengue cases to generate a hotspot. The first is likely to be strongly influenced by the degree of centrality of the subdistrict, and hence, the degree of viral import, whereas the second will additionally reflect the environmental vulnerability. With this in mind, the spatial analyses underline the degree of centrality, whereas the socio-economic analyses have generated Bangkok-wide risk factors. The extent to which centrality interferes with interpretation of the socio-economic factors is difficult to judge, but inclusion of the transport network information significantly improved the power of the SE variables to explain variation in dengue incidence rate. Significant and consistent hotspots were identified in the city centre of Bangkok. This can be attributed to several factors such as high population density, increased human activity, and high diurnal human mobility. Indeed, transport infrastructure density was significantly higher in hotspots than cold spots, suggesting connectivity is playing a significant role in disease spread. This finding is coherent with a previous study in Guangzhou city, China, where high road density was found to be a risk factor for clinical dengue [34].

This study has several strengths. Firstly, the temporal association of the meteorological variables was tested with a large dataset (2000–2013) comprising 124,820 dengue cases. The sample size was also high at over 25,000 for the main study period (2012–2013) for assessing the impact of socio-economic factors. The association was tested at the subdistrict level, which is the lowest administrative unit in Bangkok. This enabled a fine-scale spatial cluster analysis that could then be coupled with globally important SE variables. Finally, the addition of transport network data at this spatial scale revealed the importance of including such data for interpretation of SE and spatial risk factors. In future work, the connectivity and the centralities of the public transport system could be analysed using other methods (e.g., graph theory).

There were also limitations in this study. The first limitation is the fact that surveillance data are subjected to bias of under/over reporting as they only cover clinically apparent cases, whereas subclinical infections are considered the majority [55]. The fact that this study used aggregated variables at subdistrict level does not account for individual risk factors. Hence, interpretation is to be done with caution [34]. In addition, some socio-economic variables had a very broad definition comprising more than two elements, which made it hard to make a clear distinction between their roles; for example, agriculture, forestry, and fishing were included under a single category. Hence, individual/household level data collected purposefully for this analysis could have shown clearer association of these variables. Finally, the public transport system network used in this study does not account for all urban mobilities. Additional data could be used to assess daily commuting (e.g., through Call Detail Records (CDRs) or social media [56,57]).

Very few studies have considered joint effects of socio-economic, meteorological, and transport variables together and remains a field of further research [34,58,59]. More studies are needed to better understand the joint dynamics of these variables in differing urban settings whose urban structure, in terms of socio-economic and transport heterogeneity, is likely to differ. Spatial analysis has been applied frequently in understanding dengue risk in the past decade, but application of these findings is still lacking [54,60,61,62]. At the time where the disease is rapidly flourishing at the cost of limited availability of preventive measures, it is crucial to make effective and efficient use of vector control measures that are already in place. In a country such as Thailand where outbreaks occur every few years despite control efforts, there is a clear need to improve implementation of disease control strategies and identify alternative strategies. Targeting persistent hotspots may offer one such approach, but targeting the infection source may offer a more long-term solution if socio-economic factors are at the heart of the problem. An improved understanding of socio-economic vulnerabilities could help in disease prevention by directing efforts towards minimizing such attributes through allocation of resources and introducing responsive policies. However, improving our grasp of the role played by transport networks in the spread of pathogens is essential to be able to trace the sources of infection, is a subject of keen interest at all spatial scales, but currently not sufficiently developed [63,64,65,66].

While dengue outbreaks are becoming an increasing burden on the health systems of cities in the Global South, our study has important public health implications. Looking at the case of Bangkok for the years 2012 and 2013, we were able to better understand the impact of socio-economic risk factors in the distribution of spatial hotspots of dengue by adding transport variables to socio-economic and meteorological ones. The incorporation of human mobility to the analysis of infectious diseases outbreaks at city, regional, and national levels have helped improve forecasting models and early warning systems. It is often done by returning to mobile phone data, call detail records, and location data produced on social media platforms. This raises several issues of data availability, cost, and privacy protection, particularly in the context of cities and countries of the Global South.

## 5. Conclusions

The conclusions of this work are that whilst SE risk factors can be identified and thus highlight potential improvements for public health strategies (i.e., education), the inclusion of transport networks can drastically alter the outcome of risk factor analyses and thus need to be considered in further studies. Incorporating publicly available data on public transport networks proves to be robust enough to improve our understanding of the intra-urban spatial distribution of dengue outbreaks in Bangkok. A better comprehension of the role of connectivity and centrality of each locality within large conurbations could improve the implementation of disease and vector control strategies against dengue. If sources of infection and super-spreading in highly connected localities can be identified, vector control strategies could be much more focussed on such areas and alleviate the financial and manpower burden of current approaches. 

## Figures and Tables

**Figure 1 ijerph-19-10123-f001:**
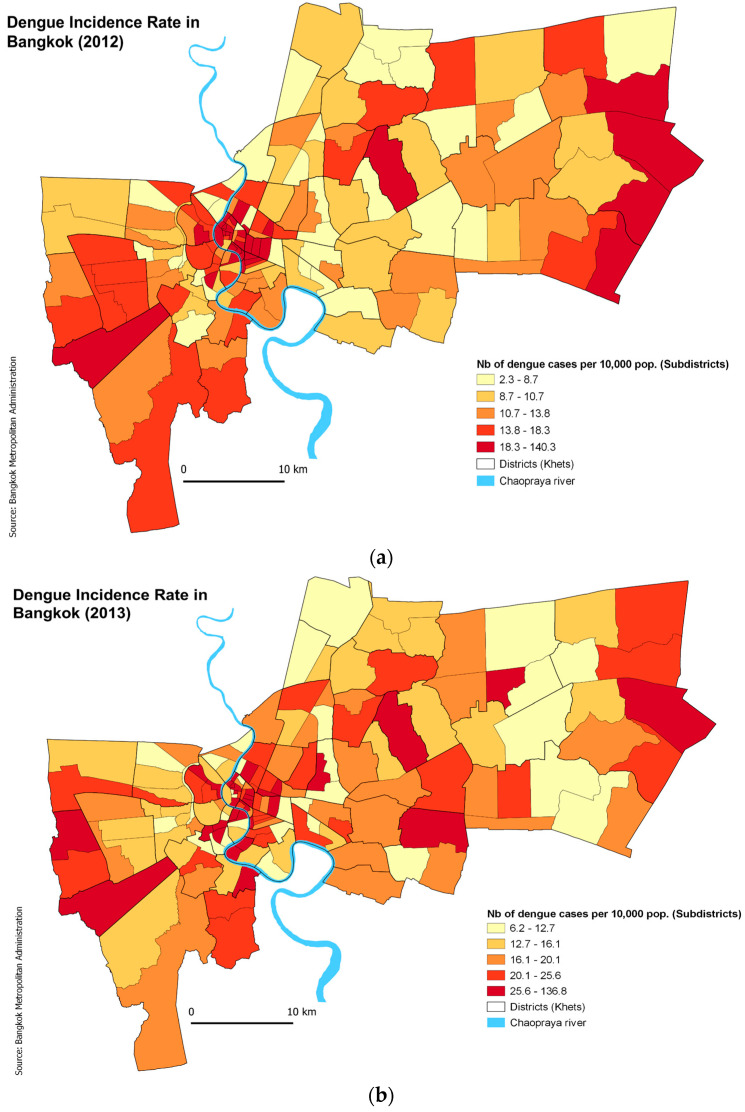
Incidence rate of dengue cases across the subdistricts of Bangkok in (**a**) 2012 and (**b**) 2013.

**Figure 2 ijerph-19-10123-f002:**
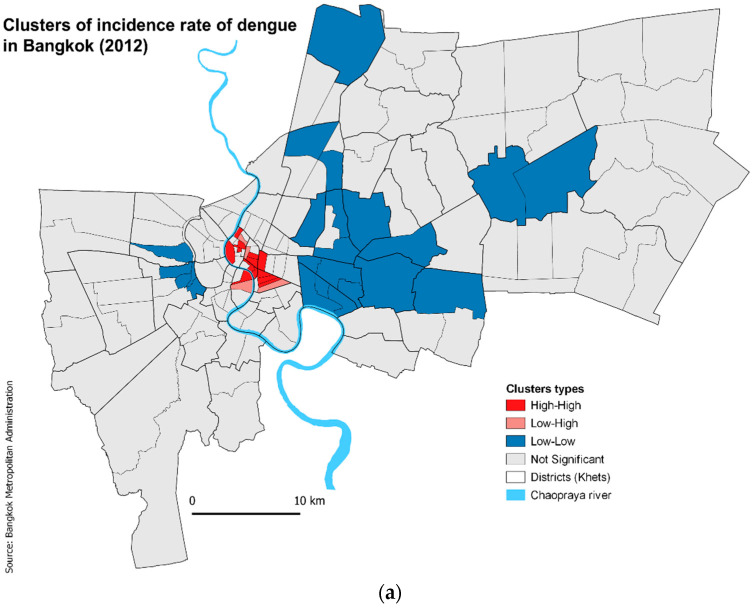
Cluster analysis of incidence rates of dengue across subdistricts in Bangkok, Thailand by year (**a**) 2012 and (**b**) 2013.

**Figure 3 ijerph-19-10123-f003:**
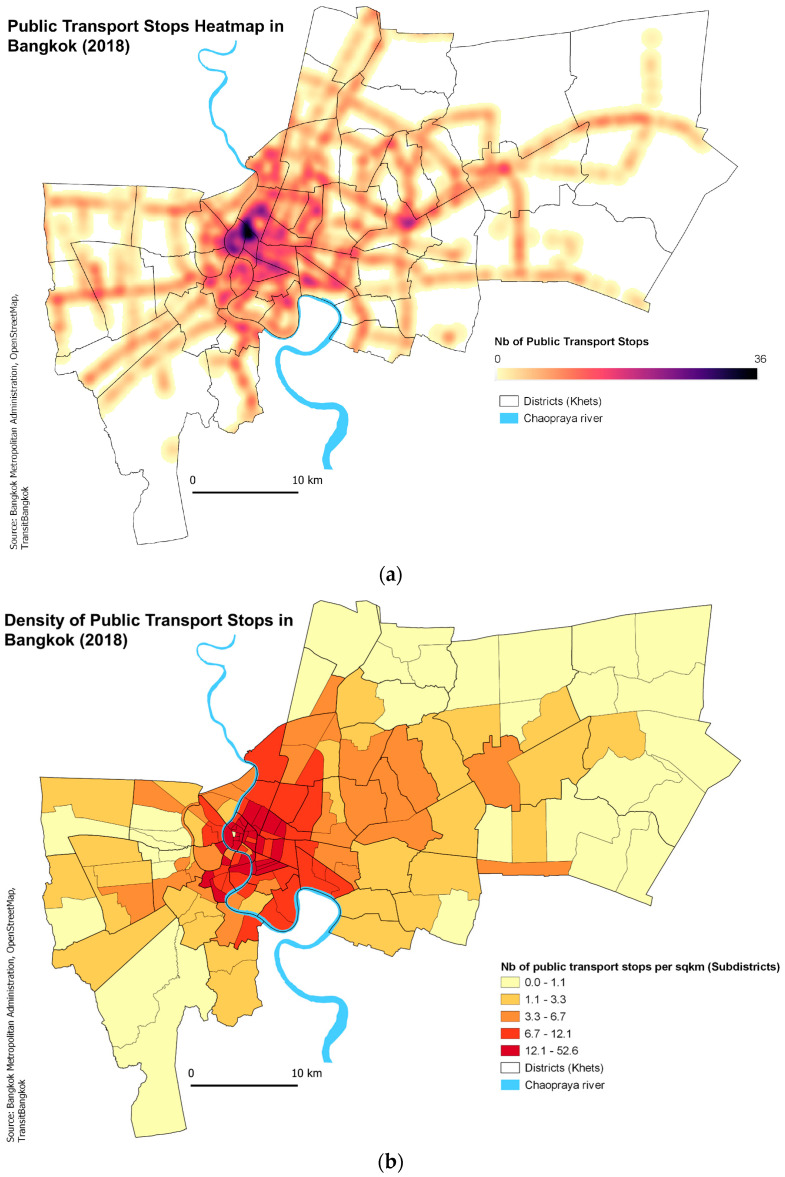
Bangkok Public Transport Network: (**a**) Public Transport Stops Heatmap, (**b**) Density of Public Transport Stops across subdistricts.

**Table 1 ijerph-19-10123-t001:** Final multivariable model for association of meteorological variables with dengue cases 2000–2013 in the final adequate model.

Variables	aRR (95% CI)	*p* Value
Year	1.03 (1.01–1.053)	0.0043
DTR (Lag 1 month)	0.81 (0.74–0.90)	<0.001
Maximum temperature	0.92 (0.85–0.99)	0.032
Mean precipitation (Lag 1 month)	1.05 (1.03–1.09)	<0.001

aRR—adjusted Relative Risk. DTR—Diurnal temperature range.

**Table 2 ijerph-19-10123-t002:** Variation in socio-economic variables among subdistricts in Bangkok.

Variable	Mean	SD
**Demography**	**(%) or N**	**(%)**
Age 0–4 years	3.77	1.65
Age 5–14 years	9.22	2.97
Age 15–24 years	16.58	4.46
Age 25–59 years	59.17	4.62
Age above 60	11.27	3.90
Number of households (N 100s)	179.3	186.3
Area (km^2^)	9.89	11.95
Built-up Area (km^2^)	0.38	0.40
Dense vegetation area (km^2^)	0.97	2.01
Low vegetation area (km^2^)	3.49	5.62
Road area (km^2^)	0.38	0.40
Waterbody area (km^2^)	0.81	3.89
Population per household	3.2	1.02
Population density (km^2^)	13,393	6324
**Education**		
No education	4.82	2.17
Primary	16.94	9.02
Secondary	16.38	8.51
Undergraduate	26.01	10.12
Postgraduate	4.65	2.67
**Nationality**		
Immigrants	9.96	8.02
**Types of occupation**		
Agriculture	1.10	2.88
Manual work	21.66	10.60
**Household characteristics**		
Cement or brick houses	74.99	14.76
Wooden houses	14.11	9.85
Shop houses	29.42	22.85
Ground water	0.05	0.10
Rain water	1.27	5.87
Air conditioner	46.21	13.89
Pit toilet	2.44	2.54

**Table 3 ijerph-19-10123-t003:** Association of socio-economic variables, meteorological factors, and year with dengue cases in the final adequate multivariable model.

Fixed Term	aRR	95% Conf. Ints	Wald Statistic	*p* Value
%No education	1.04	1.01–1.08	7.76	0.006
% Cement houses	1.006	1.002–1.01	6.95	0.009
Nb houses (100s)	1.0019	1.0015–1.0023	117.32	<0.001
lag 1 DTR	0.61	0.58–0.63	772.07	<0.001
lag 1 Mean daily Rain	1.051	1.045–1.058	319.12	<0.001
Year 2013 (vs. 2012)	1.88	1.77–2.00	403.66	<0.001
Nb transport stops	1.005	1.001–1.009	6.38	0.013

DTR—Diurnal temperature range; aRR—adjusted Relative Risk; Conf. Ints—Confidence Intervals.

**Table 4 ijerph-19-10123-t004:** Association of globally significant socio-economic variables with hotspot/cold spot clusters identified by LISA.

Variables	LISA Cluster	2012			2013		
		Mean	SE	*p* Value	Mean	SE	*p* Value
% No education	High–High	7.40	0.67	<0.001	8.16	0.86	<0.001
	Low–Low	4.22	0.47	0.300	4.18	0.79	0.310
	Low–High	6.21	0.69	0.092	5.89	1.06	0.170
	High–Low				4.77	0.75	0.806
	No cluster	4.61	0.18	Ref	4.64	0.17	Ref
%Cement house	High–High	81.12	3.87	0.055	79.04	6.18	0.258
	Low–Low	74.93	3.24	0.867	68.43	3.10	0.116
	Low–High	69.29	9.79	0.501	82.20	4.06	0.311
	High–Low				81.01	5.51	0.365
	No cluster	74.52	1.33	Ref	74.84	1.32	Ref
Nb houses (100s)	High–High	34.29	6.88	0.007	40.86	10.98	0.023
	Low–Low	316.26	57.88	<0.001	164.45	58.04	0.593
	Low–High	49.42	16.12	0.165	44.37	13.73	0.076
	High–Low				185.53	59.30	0.918
	No cluster	173.12	14.58	Ref	194.17	16.77	Ref
Public Transport Stops Density	High–High	16.32	2.12	<0.001	22.51	1.44	0.001
	Low–Low	4.20	0.63	0.237	7.74	0.89	0.032
	Low–High	17.11	2.63	0.001	25.83	2.16	<0.001
	High–Low				8.42	2.63	0.255
	No cluster	6.00	0.64	Ref	6.50	0.63	Ref

Shown are the mean, Standard errors (SE) of the variables and the *p*-Value of the regression analysis. NA—not applicable; there were no High–Low clusters in 2012.

**Table 5 ijerph-19-10123-t005:** Effect of matrices of distance and transport similarity among subdistricts and mean subdistrict number of dengue cases on association of SE variables with residual dengue cases.

	Transport Matrix		Distance Matrix	
	w/o	with	w/o	with
%No education	0.047 2.2%	0.081 36.9%	0.047 2.1%	0.072 73.1%
%Cement houses	0.0035 11.5%	0.0045 36.9%	0.0034 10.8%	−0.027 0.03%
Nb houses (100s)	0.0023 46.6%	0.0023 18.9%	0.002 48.1%	0.0036 23.3%

Shown are parameter estimates and percentage of variation explained in the multivariable GLM analyses.

## Data Availability

Not applicable.

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
