# Peer review of "Importance of Public Transport Networks for Reconciling the Spatial Distribution of Dengue and the Association of Socio-Economic Factors with Dengue Risk in Bangkok, Thailand"

_ijerph, 2022, doi:10.3390/ijerph191610123_

Round 1
Reviewer 1 Report
Dear Authors,
The article makes an interesting contribution to the field of socio-economic factors of dengue risk depending on human mobility. This is a generally well-written manuscript. However, I have a few comments to the authors before the manuscript could be considered for publication.
1. The abstract is not clear, not concise, and requires correction in line with the adopted structure for the abstract of the article, for instance, the Aim of the Study, Material and Methods, Results, and Conclusions.
2. Section Materials and Methods should begin from the paragraph "Study design", for which a detailed description is needed.
3. Some parts of the article require rewriting. Information and description on methodologies, what appear in the section "Results" should be moved to the section "Material and Methods". Please, pay attention to avoid duplication.
In addition, Table A2, what contains descriptive statistics should be presented in the section "Results“, and its findings should be moved from the "Appendix" to the main text.
Please, complete the meteorological variables used, i.e. their definitions, and descriptive statistics.
4. In the statistical analysis, it is necessary to add information on the adjusted Relative Risk (aRR), its interpretation, and the level of statistical significance.
5. The description for Figure A1 (for instance, N = 14.686 on page 5) does not agree with the values in this plot, where the scale is marked up to 2000, so this should be corrected.
In addition, there is an inconsistency in the marking Figures in the manuscript, for example, on page 5 is "S1 Fig" and in the Appendix is "Figure A1". A similar problem is with tables.
6. The first paragraph in the discussion should contain a few concluding sentences describing the „main findings”.
7. On page 15, there are no conclusions from the conducted study, which should be response to the purpose of the study. The implications described in the "Conclusions" section should be in "Discussion".
8. Abbreviations and explanations to Tables 1 to 4, Tables A1, A3 and A5, and Figure A2 should be placed under the table / figure.
9. Please, highlight the changes to the revised version using a different colour and/or a different script.
Author Response
Dear Colleague, thank you for your comments. Here are our response to them.
- The abstract is not clear, not concise, and requires correction in line with the adopted structure for the abstract of the article, for instance, the Aim of the Study, Material and Methods, Results, and Conclusions.
The abstract has been altered as follows:
Dengue is the most widespread mosquito-borne viral disease of man and spreading at an alarming rate. Socio-economic inequality has long been thought to contribute to providing an environment for viral propagation. However, identifying socio-economic (SE) risk factors is confounded by intra-urban daily human mobility, with virus being ferried across cities. This study aimed to identify SE variables associated with dengue at a subdistrict level in Bangkok, analyse how they explain observed dengue hotspots and assess the impact of mobility networks on such associations. Using meteorological, dengue case, national statistics and transport databases from the Bangkok authorities, we applied statistical association and spatial analyses to identify SE variables associated with dengue and spatial hotspots and the extent to which incorporating transport data impacts on the observed associations. We identified three SE risk factors at the subdistrict level: Lack of education, % of houses being cement/brick and number of houses as being associated with increased risk of dengue. Spatial hotspots of dengue were found to occur consistently in the centre of the city, but which did not entirely have the socio-economic risk factor characteristics. Incorporation of the intra-urban transport network, however, much improved the overall statistical association of the socio-economic variables with dengue incidence and reconciled the incongruous difference between the spatial hotspots and the SE risk factors. Our study suggests that incorporating transport networks enables a more real-world analysis within urban areas and should enable improvements in the identification of risk factors.
- Section Materials and Methods should begin from the paragraph "Study design", for which a detailed description is needed.
This section has been added and the subsequent sections renumbered.
- Some parts of the article require rewriting. Information and description on methodologies, what appear in the section "Results" should be moved to the section "Material and Methods". Please, pay attention to avoid duplication.
We have moved the sentence on the best fit model of the meteorological variables from the Results section to the Methods section.
By contrast, the explanatory sentences about what the LISA analysis is for have been left in the Results section to help the reader. The same for the transport matrices to help the reader understand.
In addition, Table A2, what contains descriptive statistics should be presented in the section "Results“, and its findings should be moved from the "Appendix" to the main text.
Table A2 has been moved to the results section and the Table numbers corrected throughout.
Please, complete the meteorological variables used, i.e. their definitions, and descriptive statistics.
The description of the variables and the descriptive stats have been added in section 2.3.2 and the second paragraph of the results section.
- In the statistical analysis, it is necessary to add information on the adjusted Relative Risk (aRR), its interpretation, and the level of statistical significance.
We have added the following in the methods section to help guide readers in the interpretation of the aRRs. For the multivariable analyses, an adjusted Relative Risk (aRR) was calculated to show the direction and size of the association of the explanatory variables with the output variables. Relative Risk for the case of continuous explanatory variables is the percentage increase in dengue cases for a unit increase in the explanatory variable (e.g. 1 °C for temperature, 1 mm for precipitation, 1% for the %SE variables etc.). The aRR is the relative risk of a variable adjusted for all the other co-fitted variables in the multivariable analysis.
- The description for Figure A1 (for instance, N = 14.686 on page 5) does not agree with the values in this plot, where the scale is marked up to 2000, so this should be corrected.
The supplementary figure 1 actually shows the median and IQ range of monthly incidence not the annual incidence. The legend has been modified to state this and the reference to the figure in the text moved to the sentence before concerning the monthly incidences.
In addition, there is an inconsistency in the marking Figures in the manuscript, for example, on page 5 is "S1 Fig" and in the Appendix is "Figure A1". A similar problem is with tables.
This has been harmonized throughout.
- The first paragraph in the discussion should contain a few concluding sentences describing the „main findings”.
The following has been added:
The major findings of this work were that although three SE variables (“%No education”, “%cement house structure” and “Nb of houses (100s)”) were associated with increased dengue IR, they poorly explained the observed dengue hotspots. Moreover, inclusion of the intra-urban transport connectivity matrix among subdistricts vastly improved the explanatory power of the fitted model and radically altered the explanatory power of the three SE variables.
In more detail, in addition to the three aforementioned SE variables “%manual labour”, was found to be associated with increased incidence of dengue and “%groundwater use” was found to be protective and only during the dry season.
- On page 15, there are no conclusions from the conducted study, which should be response to the purpose of the study. The implications described in the "Conclusions" section should be in "Discussion".
This has been altered as follows.
First part of previous conclusions paragraph put into discussion as suggested.
New conclusions paragraph:
The conclusions of this work are that whilst SE risk factors can be identified and thus highlight potential improvements for public health strategies (i.e. education), the inclusion of transport networks can drastically alter the outcome of risk factor analyses and thus need to be considered in further studies. Incorporating publicly available data on public transport networks proves to be robust enough to improve our understanding of the intra-urban spatial distribution of dengue outbreaks in Bangkok. A better comprehension of the role of connectivity and centrality of each locality within large conurbations could improve the implementation of disease and vector control strategies against dengue. If sources of infection and super-spreading highly connected localities can be identified, vector control strategies could be much more focussed on such areas and alleviate the financial and manpower burden of current approaches.
- Abbreviations and explanations to Tables 1 to 4, Tables A1, A3 and A5, and Figure A2 should be placed under the table / figure.
This has been done throughout.
- Please, highlight the changes to the revised version using a different colour and/or a different script.
In Red
Reviewer 2 Report
To my knowledge, the procedures followed are correct, but the reader would welcome additional attention in explaining and clarifying the different steps.
I appreciated the originality of the approach, and I am convinced that the conclusions could be helpful for decision-makers to adjust the prevention of the spread of mosquitoes in the urban area and, consequently, of the dengue virus.
Author Response
Dear Colleague, thank you for your comments. Here are our replies.
To my knowledge, the procedures followed are correct, but the reader would welcome additional attention in explaining and clarifying the different steps.
Thank you for this comment. We have made substantial clarifications as indicated in our responses to Reviewer 1.
We have added a study design subsection to clarify the methodology and the different steps followed.
We have moved the sentence on the best fit model of the meteorological variables in the Results section to the Methods section.
The description of the meteorological variables and the descriptive stats have been added in section 2.3.2 and the second paragraph of the results section.
We have added information on the adjusted Relative Risk (aRR), its interpretation, and the level of statistical significance in the methods section to help guide readers in the interpretation of the aRRs. See below:
"For the multivariable analyses, an adjusted Relative Risk (aRR) was calculated to show the direction and size of the association of the explanatory variables with the output variables. Relative Risk for the case of continuous explanatory variables is the percentage increase in dengue cases for a unit increase in the explanatory variable (e.g. 1 °C for temperature, 1 mm for precipitation, 1% for the %SE variables etc.). The aRR is the relative risk of a variable adjusted for all the other co-fitted variables in the multivariable analysis."
I appreciated the originality of the approach, and I am convinced that the conclusions could be helpful for decision-makers to adjust the prevention of the spread of mosquitoes in the urban area and, consequently, of the dengue virus.
Thank you for this comment. We have made the conclusions more evident as indicated in our response to Reviewer 1. See below:
"The conclusions of this work are that whilst SE risk factors can be identified and thus highlight potential improvements for public health strategies (i.e. education), the inclusion of transport networks can drastically alter the outcome of risk factor analyses and thus need to be considered in further studies. Incorporating publicly available data on public transport networks proves to be robust enough to improve our understanding of the intra-urban spatial distribution of dengue outbreaks in Bangkok. A better comprehension of the role of connectivity and centrality of each locality within large conurbations could improve the implementation of disease and vector control strategies against dengue. If sources of infection and super-spreading highly connected localities can be identified, vector control strategies could be much more focussed on such areas and alleviate the financial and manpower burden of current approaches. "
Reviewer 3 Report
This is arguably the best paper I can remember reviewing throughout my scientific career, and is certainly the best application I have read of a LISA hot spot analysis, which I usually don’t like seeing in papers because they have been so grossly over-used in most cases. Certainly, that is not the case here—the authors have provided a thorough and well-structured study rationale, and at nearly every step in reading this paper, the authors addressed my arising comments either in text or supplemental information. I say this because I want to emphasize that this is a truly outstanding paper and was an actual joy to read, which is usually not the case when I peer-review a paper for the first time! My comments below are purely minor, and I would be happy to accept this paper as it currently stands. The authors are to be commended for the work and thought that went into this manuscript, and have truly set a new bar for me as I peer-review. Thank you for this wonderful contribution!
Introduction
· Given the scope of the IJERPH special issue and the novelty of incorporating public transit here, I think a sentence or two outlining transportation in Bangkok could be warranted (either at the end of Intro or in Section 2.1). Having family in Bangkok I’m actually fairly familiar with the modes of public transportation and it might be worth noting that despite the heavy congestion (or indeed because of it!) public transit is very commonly used, particularly with the addition of the BTS line within the central area of Bangkok, which interestingly overlaps with where dengue clusters tended to be found. Towards this, I’m curious if there are any insecticide spraying operations or control efforts around BTS stops/locations? If not, this paper makes quite a good argument for doing so!
Methods
· Section 2.2.3: The thorough rationale/literature provided for the SE variables is very well received, thank you for including that as well as Table A1 which was very helpful. Given the ultimate importance of the lack of education variable, can the authors note within the rationale text slightly more detail about the ‘no education’ variable—for example, I suspect that this will be a fairly low percentage within Bangkok (I see it’s around 4%, which is even still higher than I expected!), but I’d be curious to see how an education variable with more levels might influence the analyses (e.g., ‘No education’, ‘Primary education’, ‘High school’, etc.)
Results
· Table 1: maybe consider changing variable names to ‘DTR (lag 1 month)’ and ‘Mean Precip (lag 1 month)’ just to improve visual clarity
· I have to wonder if it might be better to categorize the number of households and number of transport stops variables (e.g. into quartiles), since the associated aRR and CI’s are very small (Table 2). These variables remind of continuous age variables which are more often more easily interpreted as age categories. Regardless, this is fairly minor and is more of an interpretation of results, so feel free to disregard.
· It might be worth reminding the reader somewhere within results (maybe in section 3.2) that the SE analyses were carried out only for 2 years because of data availability. It’s stated well in the Methods section but on first read I tended to gloss over that and then was left wondering why subsequent analyses were performed only for 2012/13 (but then I figured it out based off the Methods description, thank you!). Even though it might seem redundant, I think it will help with the transition between the data used for section 3.1 to the data for section 3.2
· Section 3.3: the authors can consider noting that the null hypothesis of the Global Moran’s I is spatial dispersion—the interpretation of the Global Moran’s I in the final sentence, first paragraph of section 3.3 helps considerably for readers not familiar with this test, but I suggest adding something along the lines of “This suggests that in Bangkok there was statistically significant clustering of dengue in subdistricts in the given year, but at a low level, suggesting some spatial dispersion, as well.”
· Section 3.3: I think the clusters are supposed to be denoted as ‘High’ instead of ‘Hi’ (which is might be an abbreviation within GeoDA software?) It’s correct in Figure 2 at any rate, so please change within text.
· At nearly every step of reading this paper, as soon as a thought popped into my head, I found that the authors addressed it nearly immediately—in this case I’m referring to wanting to see a map of the Bangkok transit system and then Figure 3 addressed that beautifully. Well done, authors, this paper is truly a pleasure to read!
Author Response
Dear Colleague, we would like to thank you for your comments and your kind words regarding our paper. Here are our replies to your comments.
This is arguably the best paper I can remember reviewing throughout my scientific career, and is certainly the best application I have read of a LISA hot spot analysis, which I usually don’t like seeing in papers because they have been so grossly over-used in most cases. Certainly, that is not the case here—the authors have provided a thorough and well-structured study rationale, and at nearly every step in reading this paper, the authors addressed my arising comments either in text or supplemental information. I say this because I want to emphasize that this is a truly outstanding paper and was an actual joy to read, which is usually not the case when I peer-review a paper for the first time! My comments below are purely minor, and I would be happy to accept this paper as it currently stands. The authors are to be commended for the work and thought that went into this manuscript, and have truly set a new bar for me as I peer-review. Thank you for this wonderful contribution!
Thank you for your positive comments. Much appreciated.
Introduction
- Given the scope of the IJERPH special issue and the novelty of incorporating public transit here, I think a sentence or two outlining transportation in Bangkok could be warranted (either at the end of Intro or in Section 2.1). Having family in Bangkok I’m actually fairly familiar with the modes of public transportation and it might be worth noting that despite the heavy congestion (or indeed because of it!) public transit is very commonly used, particularly with the addition of the BTS line within the central area of Bangkok, which interestingly overlaps with where dengue clusters tended to be found. Towards this, I’m curious if there are any insecticide spraying operations or control efforts around BTS stops/locations? If not, this paper makes quite a good argument for doing so!
Thank you for these excellent ideas.
In the Introduction we have added:
"Within this context, understanding how intra-urban mobility in Bangkok may shape the spread of the virus is important. Bangkok has a very well developed intra-urban public transport system comprised of BTS Skytrains, MRT Subways, Airport Rail Link, buses and ferry boats that enable rapid transport throughout the city. Geographical knowledge of the structure of the transport network can provide a proxy for human intra-urban mobility and thus enable its inclusion in understanding dengue risk."
The vector control idea is a very good one. To our knowledge vector control is not done at bus/BTS stations, although we believe it is at Wats (Temple).
We have added a part on vector control in the conclusion. Thank you.
Methods
- Section 2.2.3: The thorough rationale/literature provided for the SE variables is very well received, thank you for including that as well as Table A1 which was very helpful. Given the ultimate importance of the lack of education variable, can the authors note within the rationale text slightly more detail about the ‘no education’ variable—for example, I suspect that this will be a fairly low percentage within Bangkok (I see it’s around 4%, which is even still higher than I expected!), but I’d be curious to see how an education variable with more levels might influence the analyses (e.g., ‘No education’, ‘Primary education’, ‘High school’, etc.)
This was actually looked at and we fail to remember why those categories weren’t included. They have been reinserted and the number of tests and Bonferroni corrected P value altered. Nothing significant in these educational groups.
Results
- Table 1: maybe consider changing variable names to ‘DTR (lag 1 month)’ and ‘Mean Precip (lag 1 month)’ just to improve visual clarity
Thank you. This has been modified as such.
- I have to wonder if it might be better to categorize the number of households and number of transport stops variables (e.g. into quartiles), since the associated aRR and CI’s are very small (Table 2). These variables remind of continuous age variables which are more often more easily interpreted as age categories. Regardless, this is fairly minor and is more of an interpretation of results, so feel free to disregard.
If the reviewer doesn’t mind we’d prefer to leave as is. Quartiles are rather hard to interpret.
- It might be worth reminding the reader somewhere within results (maybe in section 3.2) that the SE analyses were carried out only for 2 years because of data availability. It’s stated well in the Methods section but on first read I tended to gloss over that and then was left wondering why subsequent analyses were performed only for 2012/13 (but then I figured it out based off the Methods description, thank you!). Even though it might seem redundant, I think it will help with the transition between the data used for section 3.1 to the data for section 3.2
Thank you. This (2012-13) has been added at the start of section 3.2.
- Section 3.3: the authors can consider noting that the null hypothesis of the Global Moran’s I is spatial dispersion—the interpretation of the Global Moran’s I in the final sentence, first paragraph of section 3.3 helps considerably for readers not familiar with this test, but I suggest adding something along the lines of “This suggests that in Bangkok there was statistically significant clustering of dengue in subdistricts in the given year, but at a low level, suggesting some spatial dispersion, as well.”
Inserted as suggested. Thank you.
- Section 3.3: I think the clusters are supposed to be denoted as ‘High’ instead of ‘Hi’ (which is might be an abbreviation within GeoDA software?) It’s correct in Figure 2 at any rate, so please change within text.
Thank you. Changed.
- At nearly every step of reading this paper, as soon as a thought popped into my head, I found that the authors addressed it nearly immediately—in this case I’m referring to wanting to see a map of the Bangkok transit system and then Figure 3 addressed that beautifully. Well done, authors, this paper is truly a pleasure to read!
Thank you so much for your kind appreciation.